# Partial and Total Replacement of Soybean Meal with Full-Fat Black Soldier Fly (*Hermetia illucens* L.) Larvae Meal in Broiler Chicken Diets: Impact on Growth Performance, Carcass Quality and Meat Quality

**DOI:** 10.3390/ani11092715

**Published:** 2021-09-17

**Authors:** Daria Murawska, Tomasz Daszkiewicz, Wiesław Sobotka, Michał Gesek, Dorota Witkowska, Paulius Matusevičius, Tadeusz Bakuła

**Affiliations:** 1Department of Commodity Science and Animal Improvement, University of Warmia and Mazury in Olsztyn, Oczapowski Street 5, 10-719 Olsztyn, Poland; 2Department of Commodity Science and Animal Raw Material Processing, University of Warmia and Mazury in Olsztyn, Oczapowski Street 5, 10-719 Olsztyn, Poland; fox@uwm.edu.pl; 3Department of Animal Nutrition and Feed Science, University of Warmia and Mazury in Olsztyn, Oczapowski Street 5, 10-719 Olsztyn, Poland; wsob@uwm.edu.pl; 4Department of Pathological Anatomy, University of Warmia and Mazury in Olsztyn, Oczapowski Street 13, 10-719 Olsztyn, Poland; michal.gesek@uwm.edu.pl; 5Department of Animal and Environmental Hygiene, University of Warmia and Mazury in Olsztyn, Oczapowski Street 5, 10-719 Olsztyn, Poland; dorota.witkowska@uwm.edu.pl; 6Department of Animal Breeding and Nutrition, Faculty of Animal Husbandry Technology, Lithuanian University of Health Sciences, Tilžės str. 18, LT-47181 Kaunas, Lithuania; Paulius.Matusevicius@lsmuni.lt; 7Department of Veterinary Prevention and Feed Hygiene, University of Warmia and Mazury in Olsztyn, Oczapowski Street 13, 10-719 Olsztyn, Poland; bakta@uwm.edu.pl

**Keywords:** broiler chicken, full-fat *Hermetia illucens* larvae meal, growth performance, carcass quality, sensory quality of meat

## Abstract

**Simple Summary:**

The requirement of valuable protein sources for a continuously growing human population and the simultaneous decrease in available areas suitable for agricultural production present a serious future global challenge. This has prompted many researchers to take action to look for alternative sources of protein—including the use of insects. Among insects, black soldier fly (*Hermetia illucens* L., HI) larvae are a promising alternative for future sustainable nutrient sources as both feed and food. This is a very promising species to be used in poultry nutrition. Unfortunately, the results of the present study reveal that the replacement of soybean meal with high levels (50%, 75% or 100%) of full-fat HI meal in the diets of broiler chickens throughout the rearing period is unfavorable. The results of this study show that broiler chickens fed diets with high inclusion levels of HI larvae meal consumed less feed than birds fed standard diets, which was reflected in lower average daily gain. The replacement of soybean meal protein with HI larvae meal in broiler chicken diets at above 50% significantly deteriorates carcass quality and the sensory quality of meat.

**Abstract:**

The aim of this study was to compare selected growth performance parameters and slaughter characteristics in broiler chickens fed diets with a different content of full-fat *Hermetia illucens* L. (HI) larvae meal. The experiment was performed on 384 male broiler chickens (Ross 308) reared to 42 d of age and assigned to four dietary treatments (HI0—control diet and diets where soybean meal protein (SBM) was replaced with HI protein in 50%, 75% and 100%, respectively). The final body weights of chickens were as follows: 3010.0 g (HI0), 2650.0 g (HI50), 2590.0 g (HI75) and 2375.0 g (H100, *p* < 0.05). The carcasses of chickens from the experimental groups contained less meat and more abdominal fat. The feed conversion ratio for the entire experimental period was similar in groups HI0, HI50 and HI75 and more desirable than in group HI100 (*p* < 0.05). The meat of broiler chickens from groups HI75 and HI100 was characterized by significantly (*p* < 0.05) lower juiciness and taste intensity than the meat of birds from groups HI0 and HI50. The replacement of SBM protein with full-fat HI larvae meal in broiler diets exceeding 50% significantly compromised the growth performance of birds and the carcass and meat quality.

## 1. Introduction

Poultry meat is popular and widely available, and a steady increase in its production and consumption has been observed around the world [1]. Human population growth, accompanied by global climate change and a decline in the area suitable for food crops, has promoted a search for alternative, effective protein sources for livestock diets. Attention has been paid to edible insects, which are a natural source of protein and other nutrients for omnivorous species of poultry in the wild [2]. According to Gariglio et al. [3], the use of insect products in poultry diets could reduce the negative environmental impacts of poultry production.

Black soldier fly (*Hermetia illucens* L., HI) has been widely used as an ingredient of animal diets, since its larvae are a rich source of fat (7–39% dry matter—DM) and protein (37–63% DM) with a better amino acid profile than soybean meal—SBM [4]. The use of HI as a dietary protein source for animals has been investigated in rabbits [5], fish [6], pigs [7] and different poultry species. Full-fat and partially defatted HI larvae meal and HI larvae oil have been fed to Muscovy ducks [3], broiler quails [8], laying hens [9], turkeys [10,11], broiler chickens [12,13,14] and barbary partridge [15]. Partially defatted HI larvae meal incorporated into broiler chicken diets at 5% and 15% exerted a beneficial influence on the growth performance of birds and meat quality [12,13]. The inclusion of HI at 25 and 50% in the place of soybean meal positively affects the technological albumen properties [16], intestinal morphometry and enzymatic and microbial activity of laying hens [17]. The total replacement of SBM with a full fat HI larvae meal resulted in higher caecal production of butyric acid [18] and modulated the gut microbiota composition of hens, acting as a prebiotic [19]. Full-fat HI larvae meal used as a substitute for 25% SBM in laying hen diets did not compromise productive performance or egg quality [20]. Heuel et al. [21] demonstrated that SBM-based feeds can be completely replaced with black soldier fly meal and fat in the diets of high-performing layers. Experiments involving Barbary partridges revealed that the dietary inclusion of HI larvae meal had no adverse effects on carcass quality or meat quality, except for changes in the fatty acid profile [15,22]. HI larvae contain a greater amount of lipids (15% to 49%), which can be isolated [23]. HI larvae fat included as a partial (50%) or total (100%) replacement of soybean oil in diets for broiler chickens from one day of age until slaughter (35 d) had a positive effect on productive performance, carcass traits and overall meat quality, and the only possible drawback was the fatty acid profile of meat [24,25]. According to Cullere et al. [26], the replacement of soybean oil with HI larvae fat in finisher diets for broiler chickens is technically feasible up to the 100% substitution level. Chicken meat quality was satisfactory from the nutritional and sensory points of view, thus suggesting that HI larvae fat can be considered as an ingredient of commercial diets for broiler chickens. The complete replacement of SBM protein with black soldier fly protein has been successfully investigated in laying hens [18], but not in broiler chickens, which prompted us to undertake this research.

The aim of the present study was to evaluate the effect of partial (50%, 75%) or total (100%) replacement of SBM with full-fat HI larvae meal in broiler chicken diets on the growth performance of birds, carcass quality and selected meat characteristics.

## 2. Materials and Methods

According to Polish law and EU Directive 2010/63/EU, this experiment did not require the approval of the Local Ethical Committee for Experiments on Animals (Decision of the Ethics Committee: Ethic Committee Name: Local Ethics Committee, Approval Code: LKE.065.22.2019, Approval Date: 29 April 2019).

### 2.1. Birds and Husbandry

The experiment was performed in the poultry house of the Department of Commodity Science and Animal Improvement, University of Warmia and Mazury in Olsztyn (Poland). The poultry house is equipped with a waterproof floor and walls, and an automatic ventilation system. The building has a tile roof. A total of 384 male broiler chickens (Ross 308) were reared from 1 to 42 d of age, and were assigned to four dietary treatments (8 pens/treatment and 12 birds/pen). Each pen measured 1.10 m × 1.25 m (wide × long), and was equipped with a feeder, an automatic drinker and straw pellets as bedding. During the first three weeks, the birds were heated with infrared lamps to maintain a temperature consistent with standard breeding practices [27]. The lighting schedule was 18 h light:6 h dark throughout the experiment. Clinical symptoms of disease and mortality were monitored daily throughout the experiment. Feed and water were provided ad libitum throughout the trial. The diets were formulated in the Department of Animal Nutrition and Feed Science, University of Warmia and Mazury in Olsztyn (Poland). The chemical composition and energy value of full-fat *Hermetia illucens* meal and genetically modified soybean meal are presented in Table 1. The ingredients of the experimental diets are presented in Table 2.

At hatching, the chicks were vaccinated against Newcastle disease, Marek’s disease, avian infectious bronchitis and coccidiosis.

### 2.2. Formulation of Diets

The diets were formulated by including, on an as-fed basis, increasing levels of full-fat HI larvae meal (50%, 75% and 100%; HI50, HI75 and HI100, respectively). The meal was purchased from HiProMine S.A. (Str. Poznańska 8, Robakowo, Poland). In each treatment group, the diets were split into three phases: starter (1–14 d), grower (15–35 d) and finisher (36–42 d).

The ingredient composition of diets is presented in Table 2. Complete crumbled starter, grower and finisher diets for broiler chickens were prepared in the Experimental Station of Animal Nutrition in Gorzyń, Poznań University of Life Sciences. The control diet (HI0, starter, grower, finisher) contained imported, genetically modified SBM (GM-SBM) as the main vegetable protein source from high-protein feedstuffs. In experimental diets (HI50, HI75, HI100, starter, grower, finisher), SBM protein was partially and totally replaced with graded levels (50%, 75% and 100%) of HI larvae meal protein.

The ingredient chemical composition and energy value of HI and GM-SBM are presented in Table 1.

Samples of feedstuffs (HI and GM-SBM) and diets (HI0, HI50, HI75, HI100) were ground in an ultra centrifugal mill (ZM 200 Retsch, Germany) and analyzed for the content of DM, crude ash, crude protein, crude fiber and ether extract by standard methods [28]. The concentration of metabolizable energy (AME_N_) in HI larvae meal and GM-SBM was calculated based on its chemical composition and the regression equation defined in the Nutrient Requirements of Poultry [29].

The amino acid composition of HI and GM-SBM protein was determined using the AAAT 339 amino acid analyzer in hydrolyzates prepared in accordance with the manufacturer’s instructions. Tryptophan content was determined as described by [30], after sample hydrolysis with Ba(OH)_2_. The obtained data were used to determine the amino acids in experimental diets.

The content of calcium and sodium in HI larvae meal and GM-SBM was determined by atomic absorption spectrometry [31], and phosphorus content was determined by the spectrophotometric method [32].

The content of nutrients, minerals and AME_N_ per kg of the remaining feed components contained in experimental diets was estimated based on tabular data from the Nutrient Requirements of Poultry [29].

The nutritional value of experimental diets (Table 2) was calculated based on the results of our own analyses and tabular data from the Nutrient Requirements of Poultry [29]. All experimental diets met the nutrient requirements of broiler chickens presented in the Nutrient Requirements of Poultry [29].

### 2.3. Growth Performance

The health status and mortality of birds were monitored daily throughout the experiment. The live body weight (LBW) of broilers was recorded on a pen basis at the beginning of the trial (1d), and on d 14, 35 and 42. The average daily gain (ADG) and average daily feed intake (DFI) were recorded on an individual and pen basis, respectively, at the end of each growth period. The feed conversion ratio (FCR) was determined for each growth period and for the entire experiment. All measurements were performed using a high-precision electronic scale.

### 2.4. Carcass Quality

At 42 d of age, two birds per pen were selected in each feeding group based on the average final BW (±15%), and slaughtered (electrical stunning followed by cutting the jugular vein) (64 birds in total, *n* = 16). Carcasses were eviscerated after the removal of heads (between the occipital condyle and the atlas) and feet (at the carpal joint). The digestive tract, liver, heart and abdominal fat were removed. Body weight was determined before slaughter, and carcass weight was determined after bleeding and plucking. The weights of hot carcass, heart, liver, gizzard and abdominal fat were also determined. Carcasses were chilled at 4 °C for around 24 h, and were divided into the following cuts: breast muscles, legs and the remaining portion the carcass (neck + wings + trunk + sternum bones + skin of the breast part). The legs were subjected to detailed dissection (muscles + bones + skin). The weights of cuts, muscles, giblets and abdominal fat were calculated relative (%) to average BW before slaughter.

During the dissection of chilled (24 h, 4 °C) carcasses, breast muscles (*Pectoralis major—PM*, 1 bird per pen, *n* = 8) were cut out, packaged in string polyethylene bags and transported to the laboratory of the Department of Commodity Science and Animal Raw Material Processing, University of Warmia and Mazury in Olsztyn. In the laboratory, the muscles were stored at a temperature of 4 °C until quality assessment (48 h post mortem).

### 2.5. Meat Quality

#### 2.5.1. Meat pH

The pH value was measured in muscle homogenates (meat to redistilled water ratio of 1:1, *m*/*v*), with the use of a combination of the Polilyte Lab electrode (Hamilton Bonaduz AG, Bonaduz, Switzerland) and the 340i pH-meter with a TFK 325 temperature sensor (WTW Wissenschaftlich-Technische Werkstätten, Weilheim, Germany).

#### 2.5.2. TBARS Value

Thiobarbituric-acid-reactive substances (TBARS) assay was performed as described by Pikul et al. [33]. Absorbance was measured with the Specord^®^ 40 spectrophotometer (Analytik Jena AG, Jena, Germany) at a wavelength of 532 nm.

#### 2.5.3. Meat Color

The color parameters (L*—lightness, a*—redness, b*—yellowness, C*—chroma) of the *PM* muscle were determined in the CIELAB color space [34]. The values of L*, a* and b* were measured with the HunterLab MiniScan XE Plus spectrocolorimeter (Hunter Associates Laboratory, Reston, VA, USA). The final result was the arithmetic mean of three measurements taken at different points over the surface area of the ventral side of the left breast fillet. The value of C* was calculated from the formula: C* = (a*^2^ + b*^2^)^1/2^.

#### 2.5.4. Drip Loss and Cooking Loss

The values of drip loss and cooking loss were calculated using the methods proposed by Honikel [35].

#### 2.5.5. Shear Force

The shear force value was measured in *PM* muscle samples prepared as described by Honikel [35], using the INSTRON 5542 universal testing machine (Instron, Canton, MA, USA) fitted with a Warner-Bratzler head (500 N, speed 100 mm/min.). The final result was the arithmetic mean of measurements of the maximum shear force required to cut four cylinder-shaped meat samples (diameter—1.27 cm, height—2 cm) across the grain.

#### 2.5.6. Sensory Analysis

The sensory quality of cooked *PM* muscle samples (0.6% aqueous solution of NaCl, 96 °C, 1 h) was evaluated by five panelists who had been trained in the sensory properties of cooked chicken meat. A five-point scale (5 points—most desirable, 1 point—least desirable) was used. The panelists assessed meat cubes (1 cm × 1 cm × 1 cm) cut out from the center of each cooked sample. All sensory properties (aroma, taste, juiciness, tenderness) of samples were evaluated during one session. Redistilled water was made available to the panelists for mouth cleansing between samples.

### 2.6. Histological Analysis

During necropsy, samples of the right breast muscles (BM) and leg muscles (LM) (1 bird per pen, *n* = 6) were fixed in 10% neutralized formalin and embedded in paraffin blocks. Paraffin sections of the muscle (5 µm) were stained with hematoxylin and eosin (HE). Each section was imaged using the Panoramic MIDI scanner (3DHISTECH, Budapest, Hungary). Photographs were taken and muscle fiber diameters were measured using Panoramic Viewer software (3DHISTECH, Budapest, Hungary), and a minimum of 100 fibers were counted in each muscle section.

### 2.7. Statistical Analysis

The results were analyzed statistically using STATISTICA ver. 13.3 software (TIBCO Software Inc., Tulsa, OK, USA, 2017). The normality/non-normality of distribution was determined by the Shapiro–Wilk test. Pens were the experimental unit for growth performance evaluation (*n* = 8), and individual birds were used for the evaluation of carcass quality (*n* = 16 birds) and meat quality (*n* = 8, left breast muscles) and histological analysis (*n* = 6, breast muscles and leg muscles). The collected data were processed by chi-squared test (histological analysis) or one-way ANOVA (with linear and quadratic orthogonal contrasts). The results were presented as means and the standard error of the mean (±SEM). The statistical significance of differences between group means was estimated by Tukey’s test at *p* ≤ 0.05.

## 3. Results

### 3.1. Growth Performance

The growth performance of broiler chickens is presented in Table 3. The initial LBW of birds did not differ (*p* > 0.05) between dietary treatments. In successive growth stages, diets had a significant effect on the LBW of chickens (*p* < 0.001). At 14 d of age, the highest LBW (446.57 g) was noted in the control group (HI0), and it was significantly higher than in experimental groups (404.41 g, 389.11 g and 337.53 g in groups HI50, HI75 and HI100, respectively *p* < 0.05); no significant (*p* > 0.05) differences in LBW were found between groups HI50 and HI75. At 35 d of age, differences in the mean values of LBW were observed across all dietary treatments; the LBW of birds decreased (*p* < 0.05) with increasing dietary inclusion levels of full-fat HI larvae meal. At 42 d of age, LBW values did not differ (*p* > 0.05) between groups HI50 and HI75, whereas the highest LBW was noted in group HI0 (3046.00 g, *p* < 0.05), similarly to d 14 and 35. A linear downward trend in final body weight (LBW) was observed (*p <* 0.001).

Dietary treatments had a significant influence on ADG in all analyzed growth stages (Table 3, *p* < 0.001). Between d 1 and 14, ADG was highest in group HI0 (28.96 g) and lowest on group HI100 (21.09, *p* < 0.05). Similar values of ADG were found in groups HI50 and HI75 (25.92 g and 24.77 g, respectively, *p* > 0.05). Between d 14 and 35, the highest ADG (98.73 g) was noted in the control group (HI0), and it was significantly higher than in the experimental groups (85.50 g, 72.42 g and 65.36 g in groups HI50, HI75 and HI100, respectively, *p* < 0.05). Between d 35 and 42, ADG was highly similar in the control group (HI0) and in group HI50 (75.71 g and 75.72 g, respectively, *p* > 0.05); in contrast to the earlier growth stages, the highest ADG was observed in group HI100 (95.71 g, *p* < 0.05). During the periods of d 1–14 and d 14–35 in the experiment, the ADG showed a linear response (*p* > 0.001). During the period of d 35–42, ADG showed a linear (*p* = 0.035) and quadratic response (*p* = 0.012).

Daily feed intake was also significantly affected by dietary treatments (Table 3, *p* < 0.001). At 1–14 d of age, DFI was highest (37.57 g, *p* < 0.05) in the control group (HI0), and it did not differ between experimental groups (HI50—31.64 g, HI75—32.07 g, HI100—31.57 g, *p* > 0.05). A similar trend was observed at 14–35 and 35–42 d of age, with a decrease in DFI in group HI100 relative to groups HI50 and HI75 (*p* < 0.05). During the six weeks of the experiment, there was a linear decrease in daily feed intake (*p* <0.001), with the lowest value recorded in the HI100 treatment.

Dietary treatments exerted a significant effect on the FCR (Table 3). At 1–14 d of age, the FCR was lowest in group HI50 (1.239 kg/kg) and highest in group HI100 (1.491 kg/kg, *p* ≤ 0.05). Similar values of FCR were noted in the control group (HI0) and group HI75 (1.302 kg/kg and 1.300 kg/kg, respectively, *p* > 0.05). At 14–35 d of age, the FCR was comparable in groups HI0, HI50 and HI75 (*p* > 0.05) and lower than in group HI100 (*p* < 0.05). In the last feeding phase (d 35–42), the values of FCR were highest and similar in groups HI0 and HI50 (*p* > 0.05), and they were higher than those in groups HI75 and HI100 (*p* < 0.05).

During the entire experiment, the FCR was comparable in groups HI0, HI50 and HI75 (1.629, 1.586, and 1.592 kg/kg, respectively, *p* > 0.05) and more favorable than in group HI100 (*p* < 0.05). The FCR level (1–42 d) showed a linear (*p* > 0.001) and quadratic response (*p* > 0.001).

### 3.2. Carcass Quality

Dietary treatments had a significant effect on the majority of carcass quality parameters (Table 4). A linear and a quadratic response were noted for BW (*p* < 0.001 and *p* = 0.033, respectively), carcass weight (*p* < 0.001 and *p* < 0.001), breast muscles (*p* < 0.001 and *p* < 0.001), leg muscles (*p* < 0.001 and *p* = 0.002), and muscles combined (*p* < 0.001 and *p* < 0.001).

Carcass weight was significantly higher in the control group than in experimental groups receiving full-fat HI larvae meal (HI0—2239.18 g, HI50—1937.59 g, HI75—1770.53 g, HI100—1678.26 g, *p* < 0.05). Muscle weight was also highest in the control group fed with standard diets (HI0; *p <* 0.001). The weight of breast muscles was similar (*p* > 0.05) in groups HI100 (427.96 g) and HI75 (460 g), but significantly lower than in groups HI50 (538.34 g, *p* < 0.05) and HI0 (717.78 g, *p* < 0.05). The weight of leg muscles was also comparable (*p* > 0.05) in groups HI100 (385.68 g) and HI75 (460.41 g), but significantly lower than in groups HI50 (440.11 g, *p* < 0.05) and HI0 (508.97 g, *p* < 0.05). The weight of the remaining portion of the carcass was also affected by dietary treatments (*p* < 0.001) and it was significantly higher (1012.43 g) in the control group (HI0) than in experimental groups (HI50—959.14 g, HI75—911.71 g, HI100—864.62 g, *p* < 0.05). The remaining portion of the carcass showed a linear decrease (*p* < 0.001).

Increasing inclusion rates of full-fat HI larvae meal in broiler diets contributed to increased abdominal fat deposition (group *p* < 0.001 and linear response, *p* < 0.001). The weight of abdominal fat was similar in groups HI75 and HI100 (40.23 g and 42.17 g, respectively, *p* > 0.05), but significantly higher than in groups HI0 (18.77 g) and HI50 (30.84 g, *p* < 0.05). Dietary treatments had no influence on heart weight (*p* = 0.633) or gizzard weight (*p* = 0.224), but they affected liver weight (*p* < 0.001), which was significantly higher in group HI75 (60.04 g) than in the remaining groups (HI0—52.35 g, HI50—49.19 g, HI100—50.48 g, *p* < 0.05).

The applied diets exerted a significant effect on carcass yield, and the proportions of muscles, selected internal organs and abdominal fat in total BW (Table 5). Higher dietary inclusion levels of HI larvae meal led to a decrease in carcass yield (*p* < 0.001) and the percentage of breast muscles (*p* < 0.001) in the total BW of chickens (Table 5). A similar trend was observed in the combined percentage of leg and breast muscles (*p* < 0.001).

However, no significant (*p* > 0.05) differences in the proportion of breast muscles in total BW or the combined percentage of leg and breast muscles were found between groups HI75 and HI100. The proportion of leg muscles in total BW was comparable (*p* > 0.05) in groups HI0, HI50 and HI100, but significantly higher than in group HI75 (*p* < 0.05). A linear and a quadratic response were noted for carcass yield (*p* < 0.001 and *p* < 0.001) and the percentage share of muscles of body weight (breast muscles; *p* < 0.001 and *p* = 0.033, leg muscles; *p* = 0.012 and *p* = 0.046, respectively).

The experimental factor affected the relative weights of the liver (*p* < 0.001), heart (*p* = 0.024) and gizzard (*p* < 0.001). The proportion of giblets in total BW increased with increasing inclusion levels of full-fat HI larvae meal in broiler diets, and similar values were noted in groups HI0 and HI50 (*p* > 0.05) and in groups HI75 and HI100 (*p* > 0.05). A linear and quadratic response was noted for liver proportion in total BW (*p* < 0.001). The proportion of heart and gizzard in total BW showed a linear response (*p* = 0.003 and *p* < 0.001, respectively).

The inclusion rates of HI larvae meal affected carcass fat content (*p* < 0.001, linear response *p* < 0.001). Chickens fed diets HI100 and HI75 had higher abdominal fat content, in comparison with birds fed diets HI0 and HI50 (Table 5, *p* < 0.05).

### 3.3. Meat Quality

An analysis of average TBARS values (Table 6) revealed that the rate of autoxidation in the breast muscles of chickens was lower (*p* < 0.05) in group HI50 than in the control group and groups HI75 and HI100 (linear response, *p* = 0.009).

Average pH values measured in breast muscles were high in all groups (Table 6). The highest pH (*p* < 0.05) was noted in the meat of broilers fed diets containing 50% and 100% HI larvae meal, and the lowest pH (*p* < 0.05) was observed in group HI75 (guadratic response, *p* = 0.003). The differences in meat acidity were reflected in drip loss values (Table 6), which were lowest in chickens characterized by the highest muscle pH values. However, the differences between group means were not significant (*p >* 0.05), and muscle weight losses due to drip were low (approx. 1–1.15%).

Muscle samples in experimental groups were characterized by lower cooking loss than muscle samples in the control group (Table 6, linear response *p <* 0.001). The difference between groups HI75 and HI100 exceeded 4.5 percentage points (*p* < 0.05).

The breast muscles of broiler chickens from experimental groups were darker in color (lower values of L*, *p* < 0.05, linear and quadratic response was noted; *p* = 0.018 and *p* = 0.011, respectively) and had a higher contribution of redness (higher values of a*) than the muscles of control group birds (Table 6). The difference in average a* values was significant (*p* < 0.05) between groups HI0 and HI75 (linear and quadratic response; *p* = 0.006 and *p* = 0.044, respectively). In these two groups, breast muscles had a higher contribution of yellowness (higher values of b*, *p* < 0.05), compared with group HI50 (Table 6). The differences in the contribution of redness and yellowness led to differences in color saturation (C*), which was higher (*p* < 0.05) in the breast muscles of chickens from groups HI0 and HI75, compared with group HI50 (Table 6). The C* level showed a quadratic response (*p* = 0.043).

A sensory analysis (Table 7) revealed no differences (*p* > 0.05) in aroma intensity or desirability between breast muscle samples collected in the control and experimental groups, but a linear downward trend in the aroma desirability was observed (*p* < 0.001).

However, the average score for aroma desirability was slightly lower in group HI100 (linear response; *p* < 0.001). The meat of broilers fed diet HI100 was also characterized by the lowest (*p* < 0.05) tenderness and the highest shear force value (quadratic response, *p* = 0.014). The highest tenderness and the lowest shear force value were noted in the meat of broilers fed diet HI75. Average scores for juiciness and taste intensity were lower (*p* < 0.05) in groups HI75 and HI100, compared with the control group (HI0) and group HI50 (linear decrease, *p* < 0.001). The meat of broilers receiving 75% (HI75) and 100% (HI100) HI larvae meal tended to score lower for taste desirability, but a significant (*p* < 0.05) difference was found only between groups HI50 and HI75.

### 3.4. Histological Analysis

The results of histological analysis of breast and leg muscles of broiler chickens are presented in Table 8.

Segmental defragmentation and hyalinization of muscle fibers, and infiltration of lymphoid cells between fibers were the most common histopathological changes noted in breast and leg muscles in all groups. The incidence of lymphoid cell infiltration and segmental defragmentation of muscle fibers was higher in breast muscles than in leg muscles, but no significant differences were observed between groups. Hyalinization of muscle fibers in breast and leg muscles varied significantly (*p* < 0.05), and the number of lesions increased with increasing dietary inclusion levels of HI larvae meal. Two cases of fibromuscular dysplasia (FMD) in thigh muscle arteries were noted, one in group HI75 and one in group HI100. The total number of detected lesions was higher in breast muscles than in leg muscles. The diameters of breast muscle fibers (Table 8) decreased (*p <* 0.001) with increasing inclusion rates of HI larvae meal, from 62.94 µm in group HI0 to 43.99, 34.33 and 27.29 µm in groups HI50, HI75 and HI100, respectively (*p* < 0.05).

## 4. Discussion

Some authors noted improvements in the growth performance of chickens and partridges receiving HI as a diet component replacing 25% or 50% of SBM [15,36]. The results of this study show that broiler chickens fed diets with high inclusion levels of HI larvae meal consumed less feed than birds fed standard diets, which was reflected in lower ADG. Marono et al. [37] suggested that DFI could be affected by feed color, because HI meal has a dark brown color, darker than that of SBM, and therefore is less willingly consumed by birds. Some authors [23,38] suggest that another reason for lower DFI and ADG could be a high content of chitin in the diet, which is not readily digestible by monogastric animals and can negatively affect protein digestibility. In our study, the content of chitin was not analyzed, therefore it is not possible to confirm the mentioned statement. Józefiak and Engberg [39] demonstrated that the intestinal absorption of diets containing insect protein may be compromised in broiler chickens, because insects are a potential source of peptides with biological activities against gastrointestinal microbiota. In contrast to Dabbou et al. [12], who observed that the FCR was negatively affected even when defatted HI larvae meal was included in broiler chicken diets at 15%, Ognik et al. [40] found that diets containing 15% HI meal had a beneficial influence on the FCR in young turkeys. In the present study, high inclusion rates of full-fat HI larvae meal (50% and 75%) did not compromise the FCR (1–42 d, Table 3) except in group HI100, where SBM was completely replaced with HI larvae meal. Birds from groups HI50 and HI75 consumed less feed, but FCR values were comparable with those in the control group. However, the final BW of broilers decreased with increasing dietary inclusion levels of HI larvae, which could considerably compromise rearing results in intensive production systems due to less effective use of the production area. In contrast, Bovera et al. [41] found that the complete replacement of SBM with *Tenebrio molitor* larvae meal had no negative effect on the growth performance of broiler chickens.

There are no published studies investigating carcass quality and meat quality in broiler chickens or other poultry species fed diets where the substitution of full-fat HI larvae meal for SBM exceeded 50%, and therefore the discussion of the present findings is limited.

Cullere et al. [8] demonstrated that diets containing 10% and 15% HI larvae meal as a substitute for SBM and soybean oil had no negative effect on carcass weight, the weight of breast muscles or their proportion in total carcass weight in growing broiler quails. In a study by Bovera et al. [41], carcass traits were not adversely affected by the complete replacement of SBM with TM larvae meal in broiler chicken diets (from 30 to 62 d of age). In the current study, carcass quality decreased (lineary tendency) with increasing inclusion rates of HI larvae meal. The decrease in carcass yield resulted from higher proportions of components that are not intended for consumption, such as abdominal fat (linear increase), and a lower proportion of breast muscles (linear downward trend) in total BW. It should also be noted that the proportion of leg muscles in total BW was similar in the control group (HI0) and in groups HI50 and HI100, whereas the proportion of breast muscles was lower in groups HI75 and HI100, which could point to a stronger response of breast muscles to the applied dietary treatments. A different trend was observed by Popova et al. [42] in broiler chickens fed diets with a much lower inclusion level of HI meal (5%): the percentage of breast cuts increased, whereas the percentage of thigh muscles decreased significantly.

Autoxidation in the meat of broiler chickens fed HI larvae meal, expressed as TBARS value (mg of malondialdehyde (MDA) per kg of meat), has not been analyzed to date. In some experiments, soybean oil was replaced with HI larvae fat in broiler diets. Their results are inconclusive with regard to both TBARS values and the relationship between MDA content and the inclusion rate of HI larvae fat in chicken diets, which was also observed in this study of HI larvae meal. Cullere et al. [26] reported that MDA content per kg of breast meat ranged from 0.021 to 0.025 mg in broiler chickens fed diets with 50% and 100% HI larvae fat as a substitute for soybean oil, and no significant differences were found relative to the control group (0.019 mg). In contrast, Kim et al. [14] demonstrated that the meat of broiler chickens receiving diets with different inclusion levels of black soldier fly larvae oil contained significantly (*p* < 0.01) less MDA (0.12–0.13 mg/kg) than the meat of control group birds (0.24 mg/kg). It should be stressed that in the cited studies, MDA concentration in meat from broiler chickens fed diets with insect protein was far below the level of 1–2 mg MDA/kg recommended for meat and meat products [43]. This is related to the lower concentration of polyunsaturated fatty acids (PUFAs) in meat [44,45], which are susceptible to oxidation.

According to a review of research studies conducted by Kralik et al. [46], and other authors [47] the pH_24_ of broiler chicken meat varies over a relatively wide range of 5.7 to 6.2, and the most commonly cited pH value is 5.8–5.9 [43]. In the present study, average pH values ranged from 5.93 to 6.32, and were not clearly affected by the inclusion rate of HI larvae meal in broiler diets. Equally high pH_u_ values were noted by Pieterse et al. [48] and Schiavone et al. [44] in the breast muscles of broiler chickens fed control diets and diets containing HI larvae meal (6.32–6.38 and 5.98–6.04, respectively). pH_u_ values reported by Altmann et al. [49] and Popova et al. [45] were 5.80 and 5.53–5.96, respectively. Only Popova et al. [45] observed a significant effect of full-fat HI larvae meal on the average pH value, which was higher (*p* < 0.05) than the values in the control group and in the group fed partially defatted HI larvae meal.

The differences in muscle pH values between groups, noted in this study, were expected to affect the color lightness and water-holding capacity of broiler meat, since proteins far from their isoelectric points bind more water, resulting in increased WHC (lower drip loss) and, in consequence, a darker color due to greater light absorption by more hydrated meat [50]. The above relationships were observed, to a small extent, only with regard to drip loss. However, the meat of broilers receiving HI larvae meal was darker in color, with a higher contribution of redness. Somewhat lower values of L* (lightness) and b* (yellowness), and higher values of a* (redness) in the meat of broiler chickens fed HI larvae meal were also reported by Schiavone et al. [44]. According to the cited authors, higher redness could result from the accumulation of pigments from insect meal in meat, and lower yellowness could be a consequence of decreased corn gluten content in diets containing insect meal. Altmann et al. [49] and Pieterse et al. [48] found no significant differences in the physicochemical properties of meat from broiler chickens fed insect meal.

According to Cullere et al. [51], wider use of insect-based diets in livestock production and in the meat industry depends on consumers’ willingness to accept insects as feed ingredients. Potential cultural and esthetic barriers must be overcome, and health and safety concerns need to be addressed. The sensory attributes of meat derived from insect-fed animals must also be comparable with those of conventional meat products [52]. Therefore, the sensory quality of food raw materials is an important consideration. Insect meal exerts varied effects on the sensory properties of poultry meat. Onsongo et al. [53] and Pietrese et al. [49] found no significant differences in sensory scores between the breast muscles of broiler chickens fed diets with different inclusion levels of HI larvae meal. Pieterse et al. [54] and Moyo et al. [55] demonstrated that the meat of chickens fed insect protein (*Musca domestica* meal and *Imbrasia belina* meal, respectively) was characterized by a more intense metallic aroma and aftertaste, compared with the meat of control group birds. Similar observations were made by Altmann et al. [49], who also noted lower hardness and higher tenderness of meat from birds receiving HI larvae meal, compared with the control group. Gawaad and Brune [56] also found that the meat of broiler chickens fed insect protein (house fly and blow fly larvae meal) had an exceptional aroma and intense taste. In the present study, broiler meat was not evaluated for metallic flavor attributes, but the desirability of aroma and taste tended to be lower in groups HI75 and HI100. The flavor of meat from animals fed insect meal varies depending on the type (composition) of larvae rearing substrate. According to Pieterse et al. [49], further research is needed to analyze the reasons for the metallic aroma and aftertaste of meat, including iron absorption by insect larvae used for meal production. Huseynli et al. [57] identified many compounds causing the unappetizing aroma in black soldier fly larvae meal, such as butyric acid, trimethylamine, 3-methylbutanoic acid.

The quality of meat, including its sensory attributes, is affected by the fatty acid profile. As already mentioned, the meat of broiler chickens fed insect meal has lower PUFA concentrations and higher concentrations of saturated fatty acids (SFAs). In the current study, the meat of broilers fed HI larvae meal had a lower content of PUFAs (unpublished data). Polyunsaturated fatty acids play an important role in shaping the flavor of meat subjected to thermal treatment [58]. Lower concentrations of PUFAs in meat can be associated with a lower intensity of aroma and taste during sensory evaluation, which was noted in this study.

The lower juiciness of meat from broiler chickens fed diets containing 75% and 100% HI larvae meal, observed in the present study, is difficult to explain and interpret. Lower juiciness scores in the above groups were not correlated with intramuscular fat content (unpublished data), cooking loss or tenderness (which is closely related to juiciness). It appears that lower juiciness could result from the histological structure of the meat.

A histological evaluation revealed several histopathological changes, but significant differences between dietary treatments were found only in the hyalinization of muscle fibers in both breast and leg muscles; the number of lesions increased with increasing inclusion levels of HI larvae meal. The hyalinization of muscle fibers is a minor change, and it did not affect meat quality. The focal and multifocal necroses of muscle fibers, which were more frequently encountered in group HI0, are more disturbing. Similar lesions were noted in the PM muscle of heavy male broilers by Mazzoni et al. [59]. The lower number of histopathological changes in the experimental groups is related to the lower weight of breast muscles and, in consequence, the smaller muscle fiber diameter. The effect of processed animal protein (PAP) on muscle pathology has not been thoroughly investigated to date, which limits the discussion. Further research is needed to determine whether lower dietary inclusion levels of PAP would contribute to muscle pathology.

## 5. Conclusions

In summary, the present study revealed that the replacement of SBM with high levels (50–75% or 100%) of full-fat HI meal in the diets of broiler chickens throughout the rearing period is unfavorable because it compromised the growth performance of birds and the carcass quality. The meat of broiler chickens fed diets containing 75% and 100% HI larvae meal was characterized by significantly lower juiciness and taste intensity than the meat of birds from the control group and birds receiving 50% HI larvae meal. High dietary inclusion levels of full-fat HI larvae meal in broiler chicken diets had a negative effect not only on the proportion of breast muscles in total BW, but also on muscle fiber diameter. The results of this study indicate that the inclusion rate of HI protein in broiler chicken diets, used as a substitute for SBM meal, should not exceed 50%.

## Figures and Tables

**Table 1 animals-11-02715-t001:** Chemical composition and energy value of full-fat *Hermetia illucens* meal (HI) and genetically modified soybean meal (GM-SBM).

Ingredients, g/kg	HI	GM-SBM
Dry matter	931.7	895.4
Crude ash	69.4	62.5
Calcium	13.8	3.5
Total phosphorus	7.8	4.8
Sodium	2.3	0.4
Crude protein	407.6	461.6
Ether extract	293.8	18.1
Crude fibre	66.5	65.1
Gross energy (MJ/kg)	24.81	15.58
^1^ AME_N_ (MJ/kg)	18.16	9.98

^1^ Apparent metabolizable energy balanced to a zero nitrogen balance according to Nutrient Requirements of Poultry [28,29].

**Table 2 animals-11-02715-t002:** Ingredients and nutritional value of broiler chicken diets with and without full-fat *Hermetia illucens* (HI) larvae meal.

Item	Diets ^1^											
	Starter (1–14 d)				Grower (15–35 d)				Finisher (36–42 d)			
	HI0	HI50	HI75	HI100	HI0	HI50	HI75	HI100	HI0	HI50	HI75	HI100
Substitution of GM-SBM protein, % ^1^ Ingredients, g/kg	0.00	50.00	75.00	100.00	0.00	50.00	75.00	100.00	0.00	50.00	75.00	100.00
Maize	300.0	300.0	300.0	300.0	300.0	300.0	300.0	300.0	200.0	200.0	200.0	200.0
Wheat	288.0	303.6	288.1	274.6	315.6	330.3	319.0	311.2	467.6	494.6	489.6	484.1
GM-SBM	340.0	170.0	85.0	0.0	300.0	150.0	75.0	0.0	240.0	120.0	60.0	0.0
*Hermetia Illucens* larvae meal	0.0	200.0	300.0	400.0	0.0	170.0	250.0	340.0	0.0	130.0	200.0	270.0
Soybean oil	30.0	0.0	0.0	0.0	45.0	25.0	30.0	25.0	55.0	35.0	30.0	25.0
Monocalcium phosphate	15.0	4.0	4.0	3.0	13.0	4.0	4.0	3.0	12.0	3.0	3.0	3.0
Limestone	12.5	6.0	6.0	4.0	12.5	6.0	6.0	4.0	12.5	4.0	4.0	4.0
Sodium bicarbonate	1.5	1.5	1.5	1.5	1.5	1.5	1.5	1.5	1.5	1.5	1.5	1.5
Fodder salt	2.0	0.5	0.5	0.5	2.0	1.0	0.0	0.0	2.0	0.5	0.0	0.0
L-Lysine HCL (78%)	2.5	4.2	4.5	6.0	2.5	3.5	5.7	6.5	1.8	3.0	3.5	4.0
DL-Methionine (99%)	2.4	2.8	3.0	3.0	1.6	1.7	1.9	1.9	1.2	1.5	1.5	1.5
L-Threonine (99%)	0.2	1.5	1.5	1.5	0.4	1.0	1.0	1.0	0.5	1.0	1.0	1.0
Choline chloride	0.9	0.9	0.9	0.9	0.9	0.9	0.9	0.9	0.9	0.9	0.9	0.9
Mineral-vitamin premix ^2^	5.0	5.0	5.0	5.0	5.0	5.0	5.0	5.0	5.0	5.0	5.0	5.0
Calculated nutritional value g/kg												
ME_N_ kcal/kg ^3^	3000	3010	3020	3030	3100	3120	3140	3160	3170	3180	3200	3220
Lysine ^3^	13.5	13.5	13.5	13.5	11.7	11.8	11.8	11.8	10.0	10.0	10.0	10.0
Met.+ Cys. ^3^	10.1	10.2	10.2	10.3	9.5	9.6	9.6	9.6	7.5	7.7	7.7	7.7
Threonine ^3^	8.2	8.2	8.4	8.4	8.2	8.4	8.4	8.4	6.8	7.0	7.0	7.1
Methionine ^3^	6.3	6.4	6.7	6.7	5.9	6.0	6.0	6.0	4.7	4.9	5.0	5.0
Tryptophan ^3^	2.3	2.3	2.3	2.3	2.2	2.1	2.2	2.1	2.1	2.0	2.0	2.0
Total calcium ^3^	10.0	10.0	9.9	9.9	9.4	9.4	9.3	9.3	8.8	8.8	8.6	8.6
Available phosphorus ^3^	4.8	4.9	4.9	4.9	4.5	4.6	4.6	4.6	4.0	4.1	4.1	4.1
Sodium ^3^	1.7	1.7	1.8	1.8	1.7	1.6	1.6	1.6	1.7	1.6	1.6	1.6
Analysed nutrients g/kg												
Crude protein ^4^	233.6	238.1	237.8	237.2	213.8	218.2	214.9	216.3	196.1	192.5	194.5	195.0
Crude fat ^4^	50.3	74.0	93.5	131.5	64.9	75.6	105.7	125.7	75.8	82.2	104.3	119.1
Crude fiber ^4^	28.8	32.4	31.3	34.2	27.8	35.3	35.6	35.8	27.8	30.5	31.7	31.2

^1^ Levels of full-fat *Hermetia illucens* (HI) larvae meal: HI0—0%, HI50—50%, HI75—75%, HI100—100%. ^2^ Premix composition: days 1–35: 2,400,000 IU of vitamin A; 600,000 IU of vitamin D_3_; 10,000 IU of vitamin E; 600 mg of vitamin K_3_; 400 mg of vitamin B_1_; 1 400 mg of vitamin B_2_; 6000 mg of niacin (B_3_); 2800 mg of pantothenic acid; 800 mg of vitamin B_6_; 5000 µg of vitamin B_12_; 30,000 µg of biotin; 80,000 mg of choline chloride; 300 mg of folic acid; 14,000 mg of iron; 20,000 mg of manganese; 2400 mg of copper; 12,000 mg of zinc; 200 mg of iodine; 80 mg of cobalt; 50 mg of selenium; 5000 mg of antioxidant; 240 g of calcium; 14,000 mg of salinomycin—coccidiostat; 36–42 days: 18,000,000 IU of vitamin A; 600,000 IU of vitamin D_3_; 7500 IU of vitamin E; 300 mg of vitamin K_3_; 300 mg of vitamin B_1_; 1000 mg of vitamin B_2_; 4000 mg of niacin (B_3_); 2800 mg of pantothenic acid; 600 mg of vitamin B_6_; 4000 µg B_12_; 24,000 µg of biotin; 80,000 mg of choline chloride; 300 mg of folic acid; 14,000 mg of iron; 20,000 mg of manganese; 2400 mg of copper; 12,000 mg of zinc; 200 mg of iodine; 80 mg of cobalt; 50 mg of selenium; 5000 mg of antioxidant; 340 g of calcium. ^3^ Calculated according to Polish Feedstuff Analysis Tables [29]. ^4^ Values analytically determined according to standard methods [28]. ME_N_—apparent metabolizable energy corrected to zero nitrogen balance.

**Table 3 animals-11-02715-t003:** Live body weight (LBW), average daily gain (ADG), daily feed intake (DFI) and the feed conversion ratio (FCR) in broiler chickens fed diets with different levels of full-fat *Hermetia illucens* (HI) larvae meal ^1^ (arithmetic mean, SEM, *n* = 8) ^2^.

Item	Age	Dietary Treatment				SEM	*p*-Value		
		HI0	HI50	HI75	HI100		Group	Linear	Quadratic
LBW, g	1 d	41.12	41.51	42.31	42.32	0.116	0.895	0.128	0.329
	14 d	446.57 ^a^	404.41 ^b^	389.11 ^b^	337.53 ^c^	18.11	<0.001	<0.001	0.393
	35 d	2524.10 ^a^	2201.03 ^b^	1906.06 ^c^	1713.04 ^d^	24.30	<0.001	<0.001	0.321
	42 d	3046.00 ^a^	2727.00 ^b^	2504.50 ^b^	2378.00 ^c^	49.80	<0.001	<0.001	0.143
AGD, g	1—14 d	28.96 ^a^	25.92 ^b^	24.77 ^b^	21.09 ^c^	0.072	<0.001	<0.001	0.429
	14—35 d	98.73 ^a^	85.50 ^b^	72.42 ^c^	65.36 ^d^	0.096	<0.001	<0.001	0.348
	35—42 d	75.71 ^a^	75.72 ^a^	84.29 ^b^	95.71 ^c^	0.082	<0.001	0.035	0.012
DFI, g	1—14 d	37.57 ^a^	31.64 ^b^	32.07 ^b^	31.57 ^b^	0.012	<0.001	<0.001	0.293
	14—35 d	144.76 ^a^	124.29 ^b^	110.19 ^b^	101.57 ^c^	0.047	<0.001	<0.001	0.168
	35—42 d	177.86 ^a^	159.86 ^b^	159.14 ^b^	142.43 ^c^	0.109	<0.001	<0.001	0.354
FCR, kg/kg	1—14 d	1.302 ^a^	1.239 ^b^	1.300 ^a^	1.491 ^c^	0.011	0.007	<0.001	<0.001
	14—35 d	1.496 ^a^	1.483 ^a^	1.550 ^a^	1.853 ^b^	0.023	0.012	<0.001	0.012
	35—42 d	2.459 ^a^	2.195 ^b^	1.860 ^c^	1.780 ^d^	0.072	0.006	<0.001	0.021
	1—42 d	1.629 ^a^	1.586 ^a^	1.592 ^a^	1.768 ^b^	0.116	0.009	<0.001	<0.001

^1^ Levels of full-fat *Hermetia illucens* (HI) larvae meal: HI0—0%, HI50—50%, HI75—75%, HI100—100%; ^2^ SEM, standard error of the mean; ^a–d^ mean values within a row (dietary treatment) not followed by a common superscript are significantly different at *p* < 0.05.

**Table 4 animals-11-02715-t004:** Body weight (BW) ^1^ and the weights of carcass, muscles, fat and selected internal organs in broiler chickens fed diets with different levels of full-fat *Hermetia illucens* (HI) ^2^ larvae meal (g, arithmetic mean, SEM ^3^; values on d 42; *n* = 16).

Item	Dietary Treatment				SEM	*p*-Value		
	HI0	HI50	HI75	HI100		Group	Linear	Quadratic
Weight (g)								
Body (BW) ^1^	3010.00 ^a^	2650.00 ^b^	2590.00 ^b^	2375.00 ^c^	39.80	<0.001	<0.001	0.033
Cold carcass	2239.18 ^a^	1937.59 ^b^	1770.53 ^c^	1678.26 ^d^	36.41	<0.001	<0.001	<0.001
Breast muscles	717.78 ^a^	538.34 ^b^	460.41 ^c^	427.96 ^c^	19.84	<0.001	<0.001	<0.001
Leg muscles	508.97 ^a^	440.11 ^b^	398.41 ^b^	385.68 ^b^	8.74	<0.001	<0.001	0.002
^4^ Muscles	1226.75 ^a^	978.45 ^a^	858.82 ^c^	813.64 ^c^	27.31	<0.001	<0.001	<0.001
Remaining portion of the carcass	1012.43 ^a^	959.14 ^b^	911.71 ^c^	864.62 ^d^	10.56	<0.001	<0.001	0.801
Liver	52.35 ^a^	49.19 ^a^	60.04 ^b^	54.48 ^a^	1.03	<0.001	0.474	0.056
Heart	13.92	12.75	13.46	13.41	0.31	0.633	0.774	0.382
Gizzard	21.23	21.85	24.76	23.72	0.68	0.224	0.088	0.535
Abdominal fat	18.77 ^a^	30.84 ^b^	40.23 ^c^	42.17 ^c^	2.25	<0.001	<0.001	0.157

^1^ BW, body weight; average body weight of birds selected for slaughter; ^2^ levels of full-fat *Hermetia illucens* (HI) larvae meal: HI0—0%, HI50—50%, HI75—75%, HI100—100%; ^3^ SEM, standard error of the mean; ^4^ muscles (combined leg and breast muscles); ^a–d^ mean values within a row (dietary treatment) not followed by a common superscript are significantly different at *p* < 0.05.

**Table 5 animals-11-02715-t005:** Carcass yield and percentage share of muscles, selected internal organs and abdominal fat (% of BW ^1^) in broiler chickens fed diets with different levels of full-fat *Hermetia illucens* (HI) ^2^ larvae meal (%, arithmetic mean, SEM ^3^; values on d 42, *n* = 16).

Item	Dietary Treatment				SEM	*p*-Value		
	HI0	HI50	HI75	HI100		Group	Linear	Quadratic
Percentage of BW ^1^ (%)								
Carcass yield	74.39 ^a^	73.12 ^b^	68.36 ^c^	70.65 ^d^	0.42	<0.001	<0.001	<0.001
Breast muscles	23.84 ^a^	20.28 ^b^	17.77 ^c^	18.01 ^c^	0.47	<0.001	<0.001	<0.001
Leg muscles	16.92 ^a^	16.62 ^a^	15.38 ^b^	16.24 ^ab^	0.16	0.043	0.012	0.046
^4^ Muscles	40.75 ^a^	36.90 ^b^	33.15 ^c^	34.24 ^c^	0.52	<0.001	<0.001	<0.001
Liver	1.74 ^a^	1.86 ^a^	2.32 ^b^	2.29 ^b^	0.04	<0.001	<0.001	0.010
HeartD	0.46 ^a^	0.48 ^a^	0.52 ^ab^	0.56 ^b^	0.01	0.024	0.003	0.593
Gizzard	0.71 ^a^	0.82 ^ab^	0.96 ^bc^	1.00 ^c^	0.03	<0.001	<0.001	0.495
Abdominal fat	0.62 ^a^	1.16 ^b^	1.54 ^bc^	1.77 ^c^	0.09	<0.001	<0.001	0.244

^1^ BW, body weight; average body weight of birds selected for slaughter; ^2^ levels of full-fat *Hermetia illucens* (HI) larvae meal: HI0—0%, HI50—50%, HI75—75%, HI100—100%; ^3^ SEM, standard error of the mean; ^4^ muscles (combined leg and breast muscles); ^a–d^ mean values within a row (dietary treatment) not followed by a common superscript are significantly different at *p* < 0.05.

**Table 6 animals-11-02715-t006:** TBARS value and the physiochemical properties of breast muscles in broiler chickens fed diets with different levels of full-fat *Hermetia illucens* (HI) larvae meal (arithmetic mean ± SEM).

Item	Dietary Treatments				SEM	*p*-Value		
	HI0	HI50	HI75	HI100		Group	Linear	Quadratic
TBARS value (mg MDA/kg)	0.25 ^b^	0.19 ^c^	0.30 ^a^	0.27 ^ab^	0.01	<0.001	0.009	0.424
pH	6.13 ^b^	6.32 ^a^	5.93 ^c^	6.32 ^a^	0.03	<0.001	0.156	0.003
L*	60.67 ^a^	56.90 ^b^	57.34 ^b^	57.68 ^b^	0.74	0.006	0.018	0.011
a*	5.95 ^b^	6.69 ^ab^	7.95 ^a^	7.15 ^ab^	0.34	0.004	0.006	0.044
b*	13.33 ^a^	10.58 ^b^	12.49 ^a^	11.63 ^ab^	0.43	<0.001	0.121	0.043
C*	14.68 ^a^	12.55 ^b^	14.83 ^a^	13.68 ^ab^	0.43	0.003	0.720	0.290
Drip loss (%)	1.15	0.95	1.14	0.97	0.32	0.359	0.434	0.897
Cooking loss (%)	30.15 ^a^	27.19 ^ab^	25.42 ^b^	25.51 ^b^	0.80	<0.001	<0.000	0.068

TBARS—thiobarbituric acid reactive substances; MDA—malondialdehyde; ^a–c^ mean values within a row (dietary treatment) not followed by a common superscript are significantly different at *p* < 0.05.

**Table 7 animals-11-02715-t007:** Shear force values and the sensory properties of breast muscles in broiler chickens fed diets with different levels of full-fat *Hermetia illucens* (HI) larvae meal (arithmetic mean ± SEM).

Item	Dietary Treatments				SEM	*p*-Value		
	HI0	HI50	HI75	HI100		Group	Linear	Quadratic
Aroma-intensity (points)	3.40	3.00	3.00	3.05	0.30	0.750	0.444	0.463
Aroma-desirability (points)	4.45	4.75	4.35	4.00	0.20	0.117	<0.001	0.136
Taste-intensity (points)	4.15 ^a^	3.80 ^a^	3.25 ^b^	3.20 ^b^	0.12	<0.001	<0.001	0.623
Taste-desirability (points)	4.50 ^ab^	4.85 ^a^	4.20 ^b^	4.30 ^ab^	0.16	0.048	0.106	0.463
Juiciness (points)	4.60 ^a^	4.35 ^a^	3.55 ^b^	3.50 ^b^	0.20	<0.001	<0.001	0.623
Tenderness (points)	4.50	4.45	4.80	4.35	0.17	0.288	0.896	0.246
Shear force (N)	18.26 ^ab^	17.33 ^ab^	15.67 ^b^	19.37 ^a^	0.86	0.041	0.681	0.014

^a–b^ mean values within a row (dietary treatment) not followed by a common superscript are significantly different at *p* < 0.05.

**Table 8 animals-11-02715-t008:** Incidence of histopathological changes (number of cases) ^1^ in breast muscles (BM) and leg muscles (LM) and muscle fiber diameter ^2^ (arithmetic mean, SEM), in broiler chickens fed diets with different levels of full-fat *Hermetia illucens* (HI) larvae meal.

Item (*n* = 6)	Breast Muscles (BM)				Leg Muscles (LM)				*p*-Value	
	Dietary Treatment				Dietary Treatment					
	HI0	HI50	HI75	HI100	HI0	HI50	HI75	HI100	BM	LM
^1^ Histopathological changes (no. of cases)										
Loss of cross-striations	1	1	0	0	2	3	2	0	0.399	0.146
Hyalinization of muscle fibers	1	4	3	6	0	2	3	3	0.003	0.089
Focal defragmentation of muscle fibers	5	4	5	6	4	4	3	3	0.365	0.876
Focal necrosis of muscle fibers and infiltration of lymphoid cells	2	1	0	0	1	0	0	0	0.169	0.406
Multifocal necrosis of muscle fibers and infiltration of lymphoid cells and heterophils	3	1	0	1	0	1	0	0	0.142	0.406
Focal infiltration of lymphoid cells between muscle fibers	4	3	3	3	1	1	3	1	0.916	0.484
Fibromuscular dysplasia (FMD)	0	0	0	0	0	0	1	1	0.399	0.399
							*p*-value		SEM	
^2^ Breast muscle fiber diameter (µm)	62.94 ^a^	43.99 ^b^	34.33 ^c^	27.29 ^d^	-	-	-	0.000	0.58	

^1^ Significant at *p* < 0.05, (chi-squared test). ^2^ Muscle fiber diameter, one-way analysis of variance and Tukey’s test, *p* < 0.05. ^a–d^ Mean values not followed by a common superscript are significantly different at *p* < 0.05.

## Data Availability

Not applicable.

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
