# Peer review of "Partial and Total Replacement of Soybean Meal with Full-Fat Black Soldier Fly (Hermetia illucens L.) Larvae Meal in Broiler Chicken Diets: Impact on Growth Performance, Carcass Quality and Meat Quality"

_animals, 2021, doi:10.3390/ani11092715_

Round 1

Reviewer 1 Report

The manuscript describes the use of a full-fat meal from black soldier fly larve on broiler diets. The topi is not particularly innovative as several authors investigated on this aspect. However, it could contribute to increase the knowledge on the topic.

Some changes must be applied.

l. 67: delete "and" and add ", barbary partridge [....]". The reference to cite has n. 17 in your manuscript.

l. 68: add "The inclusion of HI at 25 and 50 % in replacement of soybean meal positively affects technological albume properties (Secci et al., 2020), intestinal morphometry and enzymatic and microbial activity of laying hens (Moniello et al., 2019). The total replacement of SBM with a full fat HI larvae meal resulted in higher caecal production of butyric acid (Cutrignelli et al., 2018) and modulate the gut microbiota composition of hens, acting as a prebiotic (Borrelli et al. 2017)". The manuscripts are (doi):

10.1016/j.rvsc.2017.12.020

10.1038/s41598-017-16560-6

10.3390/ani9030086

10.3390/ani10010081

l. 89: ok for the Polish law but an ethical statement must be added

Table 1: put the ingredients according to a decreasing order; the amount of crude fat is very different among the diets, however, ME is exactly the same: how is it possible? Please explain in the text.

A table with chemical-nutritional characteristics of the insect meal must  be presented

l. 153: why not the standar requirements for ROSS 308?

Statistical analysis: according your experimental design, thebest way for mean comparison is the use of orthogonal contrasts (linear and quadratic at least). Please, apply this analysis and change tables, results and discussion accordingly.

Table 2: the data of HI50 and HI100 are wrong. It is impossible to verify if results and discussion are in line with the results contained in this table. Check.

l. 386: in your case, the contents of ME of the diets must be checked. It is well known that energy content is one of the main factors affecting poultry feed intake.

l. 388-390. You did not measure the chitin

Reviewer 2 Report

Comment 1: The introduction does not justify why such high fly larvae meal levels, fed during the whole grow-out period, were investigated.

Comment 2: It is not indicated if the nutritional values of the diets are expressed on a total or a digestible basis. It can be deduced from the methodology that total (not digestible) values were used. If so, the different digestibilities of nutrients in soybean meal and fly larva meal would have been a possible confounding factor.  

Comment 3: The methodology applied does not allow to isolate the influence of three possible factors causing the observed results, therefore not allowing to state that one of them is the cause:

(1) the inclusion levels of the fly larvae meal (HI), (2) the differences in digestibilities between HI and soybean meal, or (3) the imprecision of the HI nutritional values, as some were calculated not determined by lab analysis.

Comment 4: The values in Table 2 need to be checked.

Comment 5: Footnotes under the results tables should say “not followed by a common superscript.”

Comment 6: In tables 3 and 4, need to specify the units for the response variables.

Comment 7: In table 5, clarify the units for the variable “TBARS value (mg) MDA/kg).”

Comment 8: Table 1 should present separated the calculated and determined nutritional values into different sections with the corresponding headings.

Comment 9: In line 30, “HI” should be defined for the very first time used within this specific section.

Comment 10: The reference for nutritional requirements used is not a commonly known one.

Comment 11: The authors have not presented an explanation or hypothesis for the lowest fly larvae meal inclusion level, producing the highest effect on TBARS.

Round 2

Reviewer 1 Report

Thank you for changing your manuscript

Reviewer 2 Report

Comment 1: The authors need to explain why they decided not to balance the treatment diets to be isoenergetic. The way the diets were formulated made the that the increasing levels of HI across treatments increased the fat contents and energy values; therefore, lower feed intakes were expected. Consequently, the protein intake in the treatments with higher HI inclusion rates was much lower; therefore, worse performance and carcass yield were expected.

Comment 2: The text added in lines 113-114 should be placed in line 138 or where appropriate, under that subtitle.

Comment 3: Add the soybean meal composition in Table 1.

Comment 4: The text referring to the Table 1 (and Table 1 itself) should specify if the chemical composition is the one reported by the supplier, if it comes from another reference, or if it was determined. If determined, that should be detailed in Materials and Methods.

Comment 5: In Table 2, the columns should be aligned for the values to be placed under the correct treatment heading.
